# Fabrication of Ultrafine PPS Fibers with High Strength and Tenacity via Melt Electrospinning

**DOI:** 10.3390/polym11030530

**Published:** 2019-03-20

**Authors:** Zuo-Ze Fan, Hong-Wei He, Xu Yan, Ren-Hai Zhao, Yun-Ze Long, Xin Ning

**Affiliations:** 1Industrial Research Institute of Nonwovens & Technical Textiles, College of Textiles & Clothing, Qingdao University, Qingdao 266071, China; 2017021409@qdu.edu.cn (Z.-Z.F.); yanxu-925@163.com (X.Y.); chinesezrh@126.com (R.-H.Z.); 2Collaborative Innovation Center for Nanomaterials & Devices, College of Physics, Qingdao University, Qingdao 266071, China; yunze.long@163.com

**Keywords:** melt electrospinning, PPS fibers, high tensile ductility, high-performance fibers

## Abstract

Electrospinning (e-spinning) is an emerging technique to prepare ultrafine fibers. Polyphenylene sulfide (PPS) is a high-performance resin which does not dissolve in any solvent at room temperature. Commercial PPS fibers are produced mainly by meltblown or spunbonded process to give fibers ~20 μm in diameter. In this research, an in-house designed melt electrospinning device was used to fabricate ultrafine PPS fibers, and the e-spinning operation conducted under inert gas to keep PPS fibers from oxidizing. Under the optimum e-spinning conditions (3 mm of nozzle diameter, 30 kV of electrostatic voltage, and 9.5 cm of tip-to-collector distance), the as-spun fibers were less than 8.0 μm in diameter. After characterization, the resultant PPS fibers showed uniform diameter and structural stability. Compared with commercial PPS staple fibers, the obtained fibers had a cold crystallization peak and 10 times higher storage modulus, thereby offering better tensile tenacity and more than 400% elongation at break.

## 1. Introduction

Electrospinning (e-spinning) is an emerging technology for the preparation of ultrathin fibers. Due to the high specific surface area of as-spun fibers and the convenience of in situ forming non-woven mats, the electrospun (e-spun) fibers and mats have a wide range of applications, such as filtration [1,2], environmental science [3,4], medicine [5,6], energy [7], and catalysis [8]. A typical e-spinning method utilizes a polymer solution, in most cases, containing functional components, which is formed into a fiber thanks to evaporation of the solvent before the jet arrives at a collector. However, the polymer used in the e-spinning precursor solution must be soluble in some solvent, and the solvent is volatilized during e-spinning process, which hinders industrialization of the solution e-spinning or results in the cost being high when recycling the solvent. Therefore, solvent-free e-spinning techniques are being paid more and more attention. Our groups developed a series of moisture-, light-, and heat-assisted solvent-free e-spinning technologies and in situ preparation of ultrafine fibers for water-, light-, and heat-sensitive glue systems [9,10,11,12]. Another concern is solvent-free e-spinning technology, namely melt e-spinning. It is well known that most thermoplastic polymeric materials have a certain fluidity above their melting points and, industrially, long fibers or staple fibers can be produced by meltblown or spunbonded processes. Due to equipment limitations, finer industrial meltblown fibers are more than a dozen microns in diameter. For some soluble polymers, such as polyacrylonitrile (PAN), polyvinyl alcohol (PVA), etc., wet spinning can also be carried out. Some polymer resins are difficult to dissolve in any solvents at room temperature and are only melt-spun. Thus, melt e-spinning is an effective way to fabricate ultrafine fibers, such as polypropylene (PP), polylactic acid, poly(ε-caprolactone), and the like [13,14,15,16]. Compared with solution e-spinning, melt e-spinning is more efficient and benign, and as-spun fibers are easier to control, which is beneficial before their application in electrostatic direct writing or three-dimensional (3D) printing [17,18,19].

Polyphenylene sulfide (PPS) is a kind of high-performance thermoplastic resin which is insoluble in any solvent at room temperature. PPS has special properties of strong acid resistance, heat resistance, solvent resistance, and flame retardance [20,21], and its fiber has wide application in industry, such as for environmental protection filter materials [22]. Mass production of PPS staple fibers is primarily done through meltblown spinning. The process of melt e-spinning is rarely reported. Zhang et al. used C60 as fine fillers to improve PPS electrical conductivity and fabricated crude fiber of 45–85 μm by melt e-spinning, and the e-spinning conditions were not demonstrated [23]. Other polymers added could help PPS resin to form fibers, and blended ultrafine PPS/PP fibers were obtained by Li’s group via melt e-spinning [24]. Geng et al. explored the conditions of 3D printing additive manufacturing technology with PPS employed as a raw material, and the PPS printing line was as thick as 320 μm [25]. 

In this paper, an in-house designed melt e-spinning device was employed to overcome oxidation of ultrafine PPS fibers having a higher specific surface area and the e-spinning was carried out in inert gas. The PPS ultrafine e-spun fibers were successfully afforded. The effects of e-spinning conditions on the morphologies were investigated and the performance of the resultant fibers was also characterized.

## 2. Materials and Methods

### 2.1. Materials

PPS powder with a melt flow rate (MFR) of 1714 g/10 min was kindly provided by Zhejiang NHU Company Ltd. PPS staple fibers, a commercial product with 15 μm in average diameter, was also obtained from NHU Co. Ltd. (Zhejiang, China).

### 2.2. An In-House Designed Melt Electrospinning Device

Figure 1a is an illustration of the in-house designed melt electrospinning device employed in this research. Installed into the nozzle connected to a metal syringe, a copper needle with a diameter of 2 mm helps control the viscous melt flowing of PPS and forms a Taylor cone at the needle’s top when adding a high-voltage electrostatic field. The heater is a spring-like heating ring with a temperature controller. The needle and syringe were placed into a beaker with a sealed cap, which was full of inert gas (carbon dioxide (CO_2_)), to keep the resultant jet and fiber from being oxidized. The high-voltage supply device was connected to the aluminum foil as a positive collector, surrounding the beaker.

### 2.3. Melt E-Spinning of PPS

Firstly, the CO_2_ gas was ventilated into the beaker for atmospheric displacement with a flow rate of 100 mL/min, and, 15 min later, the gas flow rate was adjusted to and maintained at 5 mL/min when e-spinning, which ensured that the jet and fiber in inert CO_2_ gas were not oxidized. PPS raw material was put into the syringe and then heated to 315 °C for 20 min. Turning on the power supply (high voltage), the PPS melt flowed of the copper needle due to gravity, and formed a Taylor cone thanks to the high electrostatic voltage. Adjusting the voltage and the tip-to-collector distance (size of beaker), the melt jet was stretched from the Taylor cone and cooled into ultrathin long fibers, deposited on the collector (Figure 1b,c).

### 2.4. Characterization

The melt e-spun fibers were observed by scanning electron microscopy (SEM, Phenom Pro, Eindhoven, the Netherlands, 10 kV accelerating voltage). The structures of PPS and fibers were characterized by means of Fourier-transform infrared spectroscopy (FT-IR, Nicolet AVATAR 370DTGS, Champaign, IL, USA). The property of stress–strain was determined with a FAVIMAT Fiber Test machine (TexTechno, Mönchengladbach, Germany). Fifteen single fibers were tested for every sample with a gauge length of 10 mm under pre-tension 0.60 cN/tex (~7.6 MPa for PPS fibers), and a loading speed at 60 mm/min. The whole testing process of e-spun PPS fibers was recorded as a video clip to observe more clearly than by naked eyes, while the tensile process by hand was also recorded, as shown in Appendix A (Appendix A), respectively. The average diameter of PPS fibers was measured based on the SEM. The thermo-mechanical properties of the e-spun PPS fibers were determined using a dynamic mechanical analysis instrument (DMA Q800, TA instruments, New Castle, DE, USA). To eliminate the noise pulse test of ultrafine single fibers, 10 fibers were twisted carefully into a rope that was measured by the DMA machine, and 10 ropes were used to ensure the repeatability of the test data, shown in Appendix A (Appendix A). The thermal functionality was analyzed by a thermogravimetric analyzer (Mettler-Toledo TGA/DSC, Columbus, OH, USA) under a nitrogen atmosphere with a heating rate of 10 °C/min, and a differential scanning calorimeter (DSC) (TA Q100, TA Instruments, USC) under the same conditions.

## 3. Results and Discussion

### 3.1. Morphologies of PPS E-Spun Fibers

As shown in Figure 2, the diameter of the e-spun PPS fibers was affected strongly by the tip-to-collector distance, nozzle diameter, and e-spinning voltage. When the spinning distance changed from 4.5 cm to 9.5 cm, the average diameter of obtained fibers was reduced from 74.09 μm to 15.50 μm (Figure 2a–c,g). When the distance increased to more than 10 cm continually, the fiber was hardly afforded, which was possibly caused by a reduction in electric field intensity. The diameter of the nozzle influences the flow rate of PPS melt, which can affect fiber diameter. As shown in Figure 2d, when the nozzles with a diameter of 3 mm were used, the average diameter of the e-spun fibers was 15.97 μm, while it was 24.95 μm when the nozzles with a diameter of 4 mm were used (Figure 2h). To keep an inert atmosphere, the flow rate of CO_2_ was held at 5 mL/min; when it was raised to more than 10 mL/min, e-spinning was hindered because the gas flow removed the heat from the beaker so quickly that the Taylor cone solidified easily at the tip of nozzle.

To obtain thinner fibers, a receiving distance of 9.5 cm and a nozzle diameter of 3 mm was employed, while the e-spinning voltage was optimized from 25 to 35 kV, as shown in Figure 2d–f. The average diameters of resultant fibers were reduced to 7.69 and 10.51 μm, respectively, when voltages of 30 kV and 35 kV were employed, which was thinner than that of fibers given at 25 kV (15.97 μm) (Figure 2d–f). Although the average diameter given at 35 kV was bigger than that at 30 kV, the fibers were uniform and the diameter distribution was narrower, as shown in Figure 2i. All the effects on the morphologies of the as-spun PPS fibers were approximate to those of other melt e-spun polymer fibers, although they were obtained with different e-spinning devices [14].

### 3.2. FT-IR Charaterization and Thermal Analysis 

The PPS raw material, as-spun fibers, and commercial staple fibers were characterized by means of FT-IR, as shown in Figure 5a. The absorption band peak at 3063 cm^−1^ was assigned to the C–H stretching vibration of a benzene ring. The peaks at 1572, 1470, and 1385 cm^−1^ were attributed to benzene ring stretching in S–C_6_H_4_–S, and those at 1090 and 1072 cm^−1^ were attributed to C-S bond stretching in S–C_6_H_4_–S. The peaks at 1008 and 804 cm^−1^ were assigned to C–H bending modes, while peaks at 740, 553, and 479 cm^−1^ were attributed to ring bending [26]. The peak appearing at 804 cm^−1^ also indicates *para*-substitution of a benzene ring (S–C_6_H_4_–S). After melt e-spinning in inert gas, the as-spun fibers inherited structural information similar to that of the staple ones, as shown in the infrared spectra (Figure 3a).

PPS is a high-performance resin with good heat resistance, and the as-prepared fibers were thermally analyzed here. Figure 3b shows that there was a glass transition point (Tg) of 153.3 °C thanks to the semi-crystallized PPS raw materials supported by the manufacturer. Staple fibers got enough time to crystallize and exhibited no obvious Tg [27,28,29]. During melt e-spinning, the PPS powder was melted, and the jet cooled rapidly to form fibers. PPS chains that were not crystallized in time had a Tg of 83.8 °C, which was close to that of amorphous PPS. A cold crystallization temperature of 113.5 °C was detected for the as-spun fibers by means of DSC. After the heating and cooling procedure, the melt point of PPS fiber was close to pristine PPS and partial crystal transformation was evident, as shown by the melting point of 285 °C [30]. 

### 3.3. DMA Spectra of E-Spun Fiber

Ten samples (ropes) were made from PPS fibers for the DMA test, which exhibited good repeatability, and the average values are recorded in Figure 4. The loss tangent (tan δ), which indicates the Tg of material, was a little higher due to the influence of testing frequency of DMA compared to those obtained by means of DSC [31]. Because of the lower degree of crystallization, the melt e-spun fiber showed three peaks of loss tangent, and the lowest peak of tan δ was related to amorphous chain motion (α-relaxation Tg), which coincides with the cold crystallization peak obtained through DSC. Thanks to the higher degree of crystallinity of the commercial staple fiber, its Tg peak disappeared gradually and the intensity of loss tangent peak decreased, which also coincided with the result of DSC [27]. The storage modulus of the as-spun fibers was 10 times higher than that of the staple ones and it exhibited a decrease-to-increase process with a cold crystallization point at 113.6 °C, whereas the staple fiber had a lower storage modulus and no cold crystallization, which conformed completely to the results shown using DSC.

### 3.4. Tensile Property (Stress–Strain)

PPS staple fibers are commercial products with an average diameter of ~20 μm; they are mainly applied in industrial filtration fields, high-temperature smoke, and dust, gas, or corrosive liquid–solid separation [32]. To make fibers thinner, the PPS jet was drawn by hot wind or mechanical force during meltblown or spunbonded procedures, and was compared to the process of melt e-spinning. The fibers given by the former process cooled slowly and had a different degree of oxidization, thereby exhibiting an elastic and brittle behavior, as shown in Figure 5a. Correspondingly, the melt e-spinning process under inert atmosphere in this work afforded fibers with a higher tensile strength, and more than 400% higher elongation at break than the staple fibers. Upon increasing the tip-to-collector distance, the as-spun fibers became thinner and, at the same time, showed higher strength and elongation at break (Figure 5b). The other commercial TORCON™ PPS fiber made by Toray Group, has 4–5 cN/dtex (~509–637 MPa) of stress strength and 20–30% of strain based on its specification [33], which is close to those of NHU’s employed in this work. It could be said that the e-spun fibers show more excellence in these properties than their commercial counterparts.

## 4. Conclusions

PPS ultrafine fibers were obtained through a melt e-spinning technique in this work. Specifically, the e-spinning operation was carried out using an in-house designed melt electrospinning device under inert atmosphere, which successfully protected the as-prepared ultrafine PPS fibers from oxidization. The e-spinning conditions were optimized, and the average diameter of the as-spun fibers was less than 8.0 μm under the following optimum conditions: nozzle diameter of 3 mm, electrostatic voltage of 30 kV, and tip-to-collector distance of 9.5 cm. Characterized by means of SEM, FT-IR, DSC, DMA, and a tensile machine, the e-spun PPS fibers exhibited uniform diameter and structural stability. Compared with commercial PPS staple fibers, the e-spun fibers exhibited a cold crystallization process and 10 times higher storage modulus, thereby offering better tensile ductility and more than 400% higher elongation at break.

## Figures and Tables

**Figure 1 polymers-11-00530-f001:**
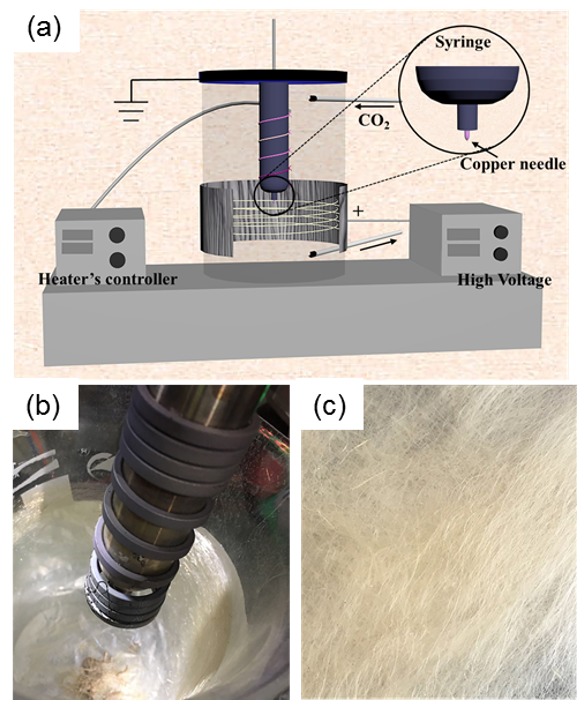
(**a**) An illustration of the melt electrospinning (e-spinning) apparatus; (**b**) optical picture of the melt e-spinning set-up; (**c**) melt e-spun polyphenylene sulfide (PPS) fibers.

**Figure 2 polymers-11-00530-f002:**
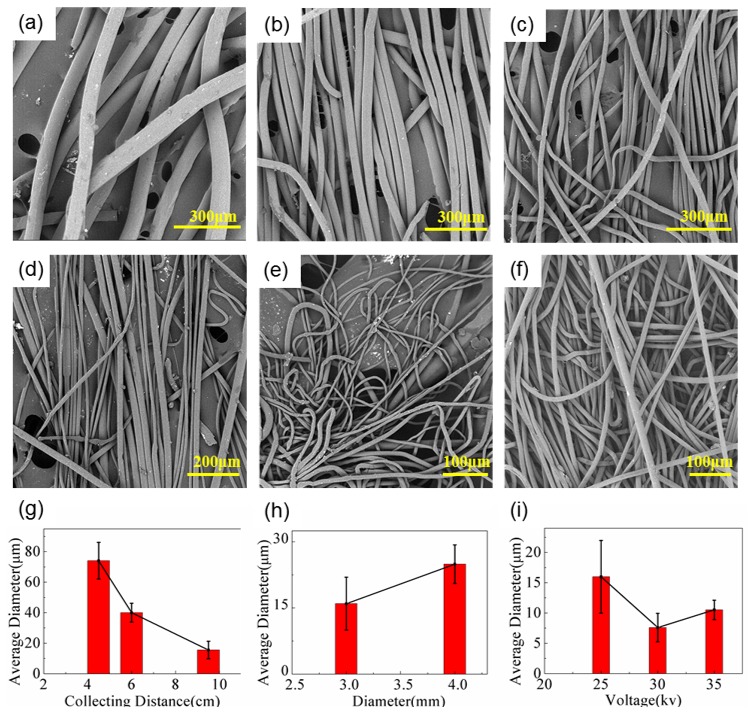
SEM images of the melt e-spun PPS fibers obtained using 25 kV of e-spinning voltage, a nozzle diameter of 4 mm, and tip-to-collector distances of (**a**) 4.5 cm, (**b**) 6 cm, and (**c**) 9.5 cm. (**d**) SEM image of e-spun PPS fibers obtained using an e-spinning voltage of 25 kV, a nozzle diameter of 3 mm diameter, and a tip-to-collector distance of 9.5 cm. SEM images of fibers obtained using a nozzle diameter of 3 mm, a tip-to-collector distance of 9.5 cm, and e-spinning voltages of (**e**) 30 kV and (**f**) 35 kV. Statistical histograms of average fiber diameter with (**g**) tip-to-collector distance, (**h**) needle diameter, and (**i**) e-spinning voltage.

**Figure 3 polymers-11-00530-f003:**
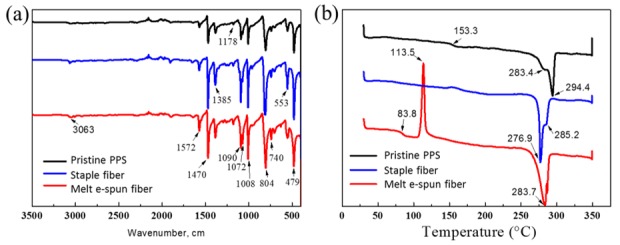
(**a**) Fourier-transform infrared (FT-IR) spectra of PPS powder (raw material) and melt e-spun PPS fibers. (**b**) Differential scanning calorimetry (DSC) heating traces at a heating rate of 10 °C/min for PPS powder and its melt e-spun fibers.

**Figure 4 polymers-11-00530-f004:**
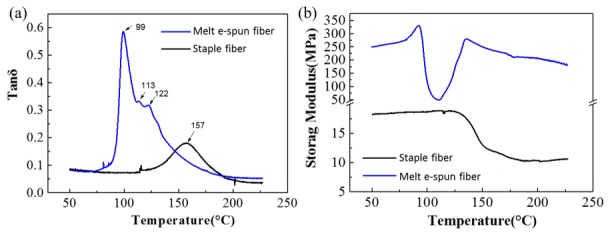
(**a**) Loss tangent (tan δ) as a function of temperature at a heating rate of 3 °C/min for PPS e-spun fibers, and (**b**) storage modulus as a function of temperature at a heating rate of 3 °C/min.

**Figure 5 polymers-11-00530-f005:**
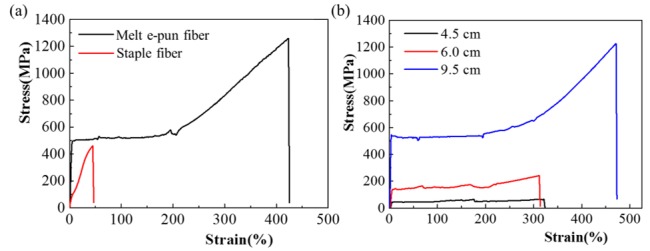
Stress–strain curves of (**a**) melt e-spun PPS fibers and commercial staple fibers, and (**b**) e-spun fibers given as a function of different tip-to-collector distances of 4.5 cm, 6.0 cm, and 9.5 cm.

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
