# Peer review of "Fabrication of Ultrafine PPS Fibers with High Strength and Tenacity via Melt Electrospinning"

_polymers, 2019, doi:10.3390/polym11030530_

Round 1
Reviewer 1 Report
In this study, a home-made melt electrospinning device was used to fabricate ultrafine PPS fibers and e-spinning operation conducted under inert gas to keep PPS fibers oxidization. The effects of e-spinning conditions on the morphologies were investigated and the performance of the resultant fibers characterized.
The paper is interesting; however the authors need to address the issues listed below before re-submission of this paper.
Major issues:
1. Explain the gas flow rate influence on the fiber diameter.
2. The section 2 “Materials and Methods” needs to be removed as this section is mainly instruction how to prepare this section.
Author Response
Dear Editors and Reviewer:
Thanks a lot for your letter and comments on our manuscript entitled “Fabrication of ultrafine PPS fibers with high strength and tenacity via melt electrospinning”. These valuable comments are very helpful for revising and improving our manuscript. We have revised the manuscript carefully in accordance with the Reviewers’ comments, which can be checked conveniently by using “tracked changes” function of Microsoft Word. And some revised parts are highlighted. The point-by-point responses to all comments are listed as follows.
The authors, once again, appreciate for the Editors/Reviewer’s work earnestly, and hope that the responses and revisions could meet with your approval.
Thanks a lot for your positive consideration.
Sincerely yours.
Zuo-Ze Fan,
Qingdao University
No. 308 Ningxia Road,
Qingdao 266071, Shandong Province, P. R. China
E-mail: 1182060882@qq.com
Tel: +86-138-6390-0519

Reviewer 2 Report
It is interesting work but I suggest few corrections
1. The most serious problem in this work – there is no discussion in this article – it must be corrected. The monologue and description of the results is not a discussion. Please compare your results with other and comment it – e.g. mechanical properties, stress –strain characteristic shapes.
You have extremely high tensile strength –5-10 times better that for some PPS commercial products:
https://www.solvay.com/sites/g/files/srpend221/files/2018-08/Ryton-PPS-Processing-
Minor corrections:
2. Please explain PPS abbreviation in abstract when it appears for the first time.
3. „home-made melt electrospinning device” -hmmm. Maybe it would be better to sound: " own design melt electrospinning device” ?
4. „(...)3D printing. [15-17] (...)”; „(...)ance, [18, 19] (...)” etc - placing links to the reference in this way is misleading, move them before the dot and commas.
5. Section 2 – please see on line 63-74 - you probably forgot to remove it from the instructions for authors.
6. Section 2: PPS meltlown fibers – do you know any properties declared by manufacturer, product name etc?
7. Section 2: Characterization: more details should be given: e.g. tensile strength - cross head speed, load cell sensitivity and range, method of fiber diameter measurements for tensile strength TS evaluation.
8. Line 207-209 – should be removed.
Author Response
Dear Editors and Reviewers:
Thanks a lot for your letter and comments on our manuscript entitled “Fabrication of ultrafine PPS fibers with high strength and tenacity via melt electrospinning”. These valuable comments are very helpful for revising and improving our manuscript. We have revised the manuscript carefully in accordance with the Reviewers’ comments, which can be checked conveniently by using “tracked changes” function of Microsoft Word. And some revised parts are highlighted. The point-by-point responses to all comments are listed as follows.
The authors, once again, appreciate for the Editors/Reviewer’s work earnestly, and hope that the responses and revisions could meet with your approval.
Thanks a lot for your positive consideration.
Sincerely yours
Zuo-Ze Fan,
Qingdao University
No. 308 Ningxia Road,
Qingdao 266071, Shandong Province, P. R. China
E-mail: 1182060882@qq.com
Tel: +86-138-6390-0519

Round 2
Reviewer 1 Report
The authors amended the manuscript as per reviewer's comments. teh amended manuscript is acceptable for publication.
Author Response
Dear Editors and Reviewers:
Thank you for the reviewers’ comments concerning our manuscript entitled “Fabrication of ultrafine PPS fibers with high strength and tenacity via melt electrospinning”. Those comments are all valuable and very helpful for revising and improving our paper, as well as the important guiding significance to our researches.
thankyou for your technical review again
Sincerely
Zuo-ze Fan
Reviewer 2 Report
It is ok. The manuscript is corrected.
Author Response

(The authors gave the same response as above.)
